# Glutathione and Glutathione-Like Sequences of Opioid and Aminergic Receptors Bind Ascorbic Acid, Adrenergic and Opioid Drugs Mediating Antioxidant Function: Relevance for Anesthesia and Abuse

**DOI:** 10.3390/ijms21176230

**Published:** 2020-08-28

**Authors:** Robert Root-Bernstein, Beth Churchill, Miah Turke

**Affiliations:** 1Department of Physiology, Michigan State University, East Lansing, MI 48824, USA; church49@msu.edu (B.C.); mturke@uchicago.edu (M.T.); 2Department of Chemistry, University of Chicago, Chicago, IL 60637, USA

**Keywords:** ascorbic acid, vitamin C, dehydroascorbic acid, glutathione, opiate, opioid, morphine, naloxone, enkephalin, methadone, epinephrine, phenylephrine, amphetamine, receptor, enhancement, synergy, antioxidant, anesthesia, addiction, drug abuse, treatment, l-cysteine

## Abstract

Opioids and their antagonists alter vitamin C metabolism. Morphine binds to glutathione (l-γ-glutamyl-l-cysteinyl-glycine), an intracellular ascorbic acid recycling molecule with a wide range of additional activities. The morphine metabolite morphinone reacts with glutathione to form a covalent adduct that is then excreted in urine. Morphine also binds to adrenergic and histaminergic receptors in their extracellular loop regions, enhancing aminergic agonist activity. The first and second extracellular loops of adrenergic and histaminergic receptors are, like glutathione, characterized by the presence of cysteines and/or methionines, and recycle ascorbic acid with similar efficiency. Conversely, adrenergic drugs bind to extracellular loops of opioid receptors, enhancing their activity. These observations suggest functional interactions among opioids and amines, their receptors, and glutathione. We therefore explored the relative binding affinities of ascorbic acid, dehydroascorbic acid, opioid and adrenergic compounds, as well as various control compounds, to glutathione and glutathione-like peptides derived from the extracellular loop regions of the human beta 2-adrenergic, dopamine D1, histamine H1, and mu opioid receptors, as well as controls. Some cysteine-containing peptides derived from these receptors do bind ascorbic acid and/or dehydroascorbic acid and the same peptides generally bind opioid compounds. Glutathione binds not only morphine but also naloxone, methadone, and methionine enkephalin. Some adrenergic drugs also bind to glutathione and glutathione-like receptor regions. These sets of interactions provide a novel basis for understanding some ways that adrenergic, opioid and antioxidant systems interact during anesthesia and drug abuse and may have utility for understanding drug interactions.

## 1. Introduction

The purpose of this study is to investigate the possibility that opioids and adrenergic compounds bind directly to glutathione and to glutathione-like regions of opioid and adrenergic receptors, accounting for some of the observed negative effects of these compounds on antioxidant functions related to anesthesia and drug abuse. Disparate sets of observations link adrenergic, opioid, ascorbate and glutathione functions through a network of interactions that include common effects on ascorbate recycling; the formation of covalently bonded drug–protein adducts involving glutathione-like cysteine-containing sequences; drug binding to highly conserved glutathione-like regions on G protein-coupled receptors; and enhancement of receptor activities by pairs of these compounds. Not only has this set of interactions not previously been noted but the mechanisms by which their interactions are manifested have not been explored fully.

Glutathione (l-γ-glutamyl-l-cysteinyl-glycine) is found in high concentrations (ca. 5 mM) in all mammalian cells and carries out a wide range of functions including acting as a coenzyme for glutathione reductase in the recycling of dehydroascorbic acid into ascorbic acid; detoxifying heavy metals, peroxides and other reactive oxygen species; catalyzing disulfide exchange reactions; maintaining disulfide bonds in proteins; and translocating amino acids across cell membranes [1,2]. Glutathione depletion, especially in the liver, commonly occurs as a side effect of the use of opioids for anesthesia and during drug abuse, resulting in the accumulation of reactive oxygen species and concomitant liver and kidney damage. This effect has been demonstrated in morphine-addicted rodents [3,4,5,6], in morphine-addicted human beings (reviewed in [5,6,7]), and the mechanism elucidated in rodent [8] and human [9,10] hepatocytes at clinically-relevant concentrations of opioids. The loss of glutathione activity extends to blood serum [10,11] and is also found in patients being treated with methadone and buprenorphine [12]. Loss of neurons associated with glutathione depletion has also been reported in rodents treated with morphine or heroin [13,14,15,16] but the evidence for equivalent depletion in human brains from opioid abusers is mixed, with one group reporting significant depletion [17,18] and another reporting none [19].

One particularly important role that glutathione plays in the liver during opioid anesthesia and drug abuse is to protect cytochrome enzymes from inactivation. Opioids are metabolized mainly in the liver through two major enzyme systems, cytochromes P450 (CYP450) (particularly CYP3A2, CPY3A4 and CYP2C8) and UDP-glucuronosyltransferases, to produce morphine 3-glucuronide and to a lesser extent, morphine 6-glucuronide, morphine 3-sulfate and normorphine [20,21]. CYP3A4 and CYP2B6 are responsible for the metabolism of methadone to its inactive form, 2-ethylidene-1,5-dimethyl-3,3-diphenylpyrrolidine [22]. The metabolism of opioids to their inactive forms results in covalent modification of one of the cysteine residues anchoring heme to the cytochromes [22,23]. Glutathione mediates the inactivation of cytochrome enzymes by competing with cysteine residues anchoring heme.

Exposure to opioids such as oxycodone results in a greater than six-fold increase in expression in rat liver of glutathione S-transferase A-5, an enzyme that catalyzes conjugation of glutathione to toxic substances, while the expression of the cytochrome enzyme CYP3A2 is decreased seven-fold [24]. The increase in glutathione S-transferase is likely due to up-regulation of its synthesis in response to its inactivation by opioids. Morphinone, a reactive electrophile, is another major product of opioid metabolism in the liver [25,26]. Morphinone binds covalently to proteins in liver specifically at cysteines to produce morphinone–cysteine adducts that interfere with protein function [27,28,29,30]. This adduct reaction is inhibited, and protein function maintained, by the presence of sulfhydryl compounds such as glutathione and L-cysteine [28,31], *S*-adenosyl-l-methionine [32], and the disulfide-containing garlic compound ajoene ((*E*)-1-(prop-2-enyldisulfanyl)-3-prop-2-enylsulfinylprop-1-ene) [33]. The mechanism of this protection appears to be a competition reaction: morphinone reacts covalently with the sulfur-containing compounds in preference to the proteins. In the case of glutathione (GSH), morphinone is inactivated by production of (8*S*)-(glutathion-*S*-yl)dihydromorphinone [34,35,36]. GSH depletion and oxidative cell damage result from the formation of the opioid–glutathione adduct interfering with ascorbic acid recycling and all other glutathione functions [37].

Codeine, which is naturally metabolized to morphine, can similarly be metabolized into morphinone but also into codeinone [38,39]. Codeinone, which can form adducts with glutathione [29,40], is toxic at high concentrations [41,42] and is, like morphinone, antagonized by the co-administration of either glutathione or L-cysteine by forming neutralizing adducts [38].

Glutathione adducts have also been characterized for the opiate agonist oxycodone and its antagonist, naltrexone [43] as well as for pharmacologically unrelated drugs such as acetaminophen [44,45], cocaine [46], methylenedioxymethamphetamine (MDMA) [47], and Δ9-tetrahydrocannabinol (THC) [43]. Thus, many classes of drugs impair glutathione function through covalent reactions that deplete its cellular availability.

Glutathione is not the only antioxidant system at work in physiological systems. An extracellular system with glutathione-like activity has recently been reported on the extracellular loops of adrenergic and histaminergic receptors [48]. Peptides derived from these extracellular loop regions that contain either cysteines or methionines bind dehydroascorbic acid and/or ascorbic acid. Some of the cysteine-containing sequences, in the presence of phosphate, convert dehydroascorbic acid it into ascorbic acid at rates similar to those observed for glutathione [48].

Additionally, the extracellular loops of the aminergic receptors containing these glutathione-like sequences are associated with the enhancement of aminergic receptor function in the presence of ascorbic acid, suggesting that binding of ascorbic acid to these glutathione-like regions results in allosteric modification of receptor function at physiologically relevant concentrations [49,50,51,52,53]. Opioids, including morphine, enkephalins and naloxone, also enhance aminergic receptor activity at physiologically relevant concentrations (e.g., [54,55,56,57,58,59,60,61,62,63,64,65], reviewed in [66]). Binding of opioids to aminergic receptors occurs at the same glutathione-like regions as does ascorbic acid [67,68]. The receptor enhancement produced by both ascorbic acid and opioid drugs is characterized by increased receptor activity in the presence of agonists at any submaximal dose, increased duration of activity at any given agonist dose, as well as reversal of tachyphylaxis and/or fade in tissues regulated by the receptor [49,50,51,52,58,59,60,61,62,63,69]. The inhibition or reversal of tachyphylaxis and fade by both ascorbate [70,71,72,73,74,75] and opioids [65,76,77] is mediated by increased cGMP production and concomitant inhibition of the G protein-mediated phosphorylation that down-regulates the receptor.

Similarly, many aminergic drugs bind to the extracellular loops of the mu opioid receptor at physiologically relevant concentrations where they function to enhance opioid binding [67,68]. Moreover, aminergic receptors dimerize with opioid receptors so that each can co-activate, or down-regulate, each other [78,79]. Thus, a complementary association of aminergic and opioid activities exists in which adrenergic compounds enhance opioid receptor function while opioids enhance adrenergic receptor function; both types of enhancement are associated with ascorbic acid binding regions of the receptors.

Three additional observations also serve to link antioxidant activity, opioids and adrenergic compounds. First, ascorbic acid binds directly to adrenergic compounds forming a stable complex that greatly retards oxidation of the amines [80,81,82,83,84]. Next, ascorbic acid may also serve to protect the disulfide bonds that stabilize the highly active forms of adrenergic and opioid receptors [85]. Finally, adrenergic compounds bind directly to opioids, which similarly retards amine oxidation [86]. These observations are consistent with the opioids sharing a common binding motif with ascorbic acid [66,87] that may mediate their shared binding to glutathione.

Because of the structural and functional similarities shared by glutathione and some peptide sequences derived from extracellular loops of aminergic receptors, we decided to compare the binding of ascorbic acid and relevant aminergic and opioid drugs to glutathione and to glutathione-like peptides derived from relevant receptors. The resulting studies may help to explain the mechanisms by which aminergic and opioid drugs interact with both sets of receptors and interfere with antioxidant activity simultaneously.

## 2. Results

Eighteen peptides (Figure 1) were tested for their ability to bind ascorbic acid, dehydroascorbic acid, adrenergic compounds, and opioid compounds as well as a number of control compounds such as glucose and various neurotransmitters. The method used was ultraviolet spectrophotometry, which is a sensitive way to test for the binding of pairs of molecules. Binding data obtained using this method have previously been validated by direct comparison to results of nuclear magnetic resonance, circular dichroism, capillary electrophoresis, and other physicochemical methods (e.g., [51,87,88,89,90]).

The opioid drugs morphine, methadone, naloxone and methionine-enkephalin (ME) bound to reduced glutathione (GSH) with Kd values of approximately 60 µM, which was the binding constant of dehydroascorbic acid (DHA) to GSH as well and significantly better than that of ascorbic acid (AA) (Figure 2; Table 1). Thus, mole for mole, opioids compete directly with DHA for GSH.

Oxidized glutathione (GSSH) has a significantly lower affinity for most of the compounds tested than reduced glutathione (GSH), presumably because the disulfide bond that forms GSSH results in decreased affinity for the sulfide side chain of the cysteine (Figure 3; Table 1 and Table 2).

Significant binding was observed between adrenergic compounds such as phenylephrine and propranolol, and even adrenergic precursors such as tyrosine and phenylalalanine, and the cysteine-containing adrenergic receptor peptide 183–185 (NCY) (Figure 4; Table 3). No binding was observed between this peptide and various control compounds such as glucose, acetylcholine, glycine, and glutamate. Notably, this peptide had no observable affinity for opioid compounds either but did bind ascorbic acid (Table 2). 

Binding of test compounds to the mu opioid receptor (muOPR) region 111–122, which lacks a cysteine residue and is therefore significantly unlike glutathione, was minimal or unobservable (Figure 5). Such minimal binding was also observed for the other peptides lacking cysteine that were tested, such as muOPR 121–131 and the insulin receptor peptides (Table 1).

In contrast to the relative lack of binding to the muOPR peptides 111–122 and 121–131, muOPR 132–143 displayed significant binding to ascorbic acid but not dehydroascorbic acid, and adrenergic compounds such as phenylephrine and L-DOPA (Figure 6; see also Table 2). Notably, this peptide also bound opioid compounds such as methadone (Figure 6) as well as morphine, naloxone and met-enkephalin (Table 2).

Each of the drugs tested in these experiments yielded a diversity of binding curves to the various receptor peptides (Table 1 and Table 2). Figure 7 provides a typical example, in this case of the binding of propranolol to peptides derived from the various classes of receptors tested here.

Overall, receptor peptides tended to fall into classes depending on their affinities for ascorbic acid, opioids and adrenergic compounds (Table 1 and Table 2). Reduced glutathione had significant affinity for ascorbic acid, opioids and adrenergic compounds as well as serotoninergic compounds, but not for other neurotransmitters or glucose. Oxidized glutathione tended to have less affinity for these compounds except for dehydroascorbic acid and some of the adrenergic compounds. Mu opioid receptor peptides that contained cysteines tended to mimic the binding profile of reduced glutathione with the exception of having little affinity for dehydroascorbic acid. Opioid receptor sequences lacking a cysteine had little affinity for any of the tested compounds, which was also typical of the insulin receptor sequences tested. Aminergic receptors tended to have one of two profiles: those that bound opioids tended to bind serotoninergic compounds but not adrenergic compounds; those that bound adrenergic compounds tended to have lower affinities for opioid compounds and to lack affinity for serotoninergic compounds. Additionally, longer peptides tended to display a higher degree of discrimination among the various ligands than did shorter compounds, suggesting that conformation becomes a discriminatory factor with increased peptide length. Notably, however, none of the peptides tested had observable affinity for glucose, glycine, acetylcholine, glutamate, or (with one histamine receptor peptide) histamine. Thus, despite the wide range of adrenergic and opioid compounds that demonstrated affinity for the adrenergic and opioid receptor peptides tested, their affinity is nonetheless class specific and similar to the affinities observed for glutathione.

## 3. Discussion

Our data suggest that a wider range of pharmacologically important interactions occur between glutathione and adrenergic, histaminergic and opioid drugs than anyone has reported before and we report here, for the first time, similar binding of these classes of drugs to glutathione-like regions of aminergic receptors that are involved in extracellular ascorbic acid recycling. These data are, however, limited to binding to isolated extracellular loop GPCR peptides, not to intact receptors, and must be interpreted with caution. We note, though, that many experiments demonstrating in vitro and in vivo effects of these drugs have been carried out (as summarized in the Introduction) and that glutathione-like activity (including recycling of ascorbic acid and the transformation of dehydroascorbic acid back into ascorbic acid) has been demonstrated for both intact GPCR and for many of the peptides tested here [48,49,50,51,52].

The results summarized in Table 1 demonstrate that ascorbic acid binds to reduced glutathione with a binding constant of approximately 60 µM and to oxidized glutathione at approximately 20 µM (with lower-affinity binding at 310 µM). Dehydroascorbic acid binds similarly. Opioids also bind with similar affinities to reduced glutathione (ca. 60 µM) but with significantly lower affinity for oxidized glutathione. Several adrenergic compounds, including phenylephrine, propranolol, amphetamine, and epinephrine, bind with even higher affinity to reduced glutathione (5 to 20 µM) as well as to oxidized glutathione. These results suggest that many of these compounds may be able to participate in the formation of the types of glutathione adducts described in the Introduction above. Other sugars (e.g., glucose) and bioactive small molecules (histamine, glycine, glutamate, etc.) do not bind appreciably to either reduced or oxidized glutathione with the exception of serotonin and to a lesser extent melatonin.

The results summarized in Table 1 further demonstrate that several of the adrenergic compounds, especially epinephrine, norepinephrine and amphetamine, have high affinity for extracellular opioid receptor peptides, with binding constants in the 1–5 µM range, but do not bind significantly to a transmembrane region (OPR 121–131). Notably, opioid binding to these peptides is significantly less than the adrenergic compounds, suggesting that the function of these extracellular regions of the opioid receptor has been optimized to facilitate allosteric control of the receptor by adrenergic compounds rather than to attract opioids to the receptor. Previous studies have demonstrated that binding of adrenergic drugs to intact OPR at these concentrations results in increased opioid binding [67,68]. Ascorbic acid has high affinity (5 µM) for one of the five OPR peptides tested (muOPR 132–143), and moderate affinity (ca. 70 µM) for another two (muOPR 38–51 and 211–226); this finding may indicate that ascorbate can also function as an allosteric modulator of the OPR, which would be consistent with the fact that it shares some common structural binding motifs with opioid compounds [66,67,68] as well as shared GPCR enhancement [49,50,51,52,58,59,60,61,62,63,65,69,70,71,72,73,74,75,76,77]. Dehydroascorbic acid and glucose had no measurable affinity for the OPR peptides nor did any of the other control compounds except, once again, serotonin and melatonin.

The results summarized in Table 2 demonstrate that ascorbic acid and dehydroascorbic acid bind to select aminergic receptor peptides, with affinities ranging from 7 to 65 µM, which is within the physiological range for these compounds in blood serum and equivalent to, or better than, their affinity for reduced and oxidized glutathione. Opioids tend to bind to the same set of adrenergic receptor peptides as ascorbic acid and with similar binding constants, suggesting a similarity between the effects of the two sets of compounds. Notably, the binding of adrenergic compounds to these aminergic receptor peptides tends to be the inverse of the ascorbic acid–opioid binding, so that peptides that bind one set of compounds do not bind the other, and vice versa. To some extent, serotonin, melatonin and histamine mimicked the behavior of the adrenergic compounds, suggesting that these extracellular regions of the aminergic receptors are fairly non-specific in their attraction for amines in general. The other control compounds did not bind to any of the aminergic receptor peptides with measurable affinities.

In sum (Table 3), the binding of opioids and adrenergic compounds to extracellular glutathione-like peptides derived from aminergic and opioid receptors generally mimics both the specificity and the affinity that they have for glutathione itself. The receptor peptides have, however, evolved to discriminate better than glutathione between opioid and adrenergic compounds, tending to be optimized for one set or the other although significant overlaps still exist. All of the receptor peptides mimic glutathione in discriminating clearly between opioids or amines and other types of bioactive small molecules such as glucose, glycine, glutamate, histamine, etc. Notably, however, serotonin and melatonin share with the adrenergic compounds significant affinity for many of the same receptor peptides and may therefore have similar effects on opioid function.

The binding studies reported here are relevant to understanding antioxidant metabolism. Ascorbic acid is present in 50–110 µM concentrations in the blood plasma of normal human beings, who also have approximately 5–20 µM dehydroascorbic acid present [91,92]. The binding constants reported here for ascorbic acid binding to receptors range from approximately 10 to 300 µM, which translates to anything from nearly complete saturation of these binding sites on the receptors to approximately one-third saturation. Dehydroascorbate has significantly less affinity for most of the receptor peptides tested so that very little of it will be bound to the receptors under normal physiological conditions. However, while ascorbic acid has not been shown to produce adducts with glutathione, dehydroascorbic acid can [93,94]. A reasonable inference is that dehydroascorbic acid may also undergo a covalent reaction with aminergic receptor loops as well, degrading receptor function over time if there is a significant amount of ascorbate oxidation occurring. This prediction is experimentally testable and of possible clinical significance in situations in which overproduction of dehydroascorbate might be expected, such as under conditions in which ascorbic acid is overwhelmed by reactive oxygen species (ROS). ROS, in other words, may indirectly antagonize aminergic and opioid receptors by producing covalent adducts that interfere with ligand binding.

Binding of bioactive amines and aminergic drugs to glutathione itself has been known since the 1960s but little studied. Studies in the 1960s [95,96] demonstrated in vitro, in both aqueous solution and also in human blood samples, that epinephrine binds directly to glutathione and when oxidized to its adrenochrome form, epinephrine can undergo a covalent reaction producing an adduct that involves the glutathione sulfhydryl group. More recently [97], it was found that such adducts are also formed in vivo under chronic high-stress conditions (and, presumably, if a patient were treated with epinephrine pharmacologically). Amphetamines also form adducts with glutathione under physiological conditions [98,99,100] as do dopamine-related catechols [101,102], which can also form adducts with other thiol-containing proteins. The damage to the glutathione system by amphetamines, which includes depletion in dopamine/serotonin nerve rich areas of the brain [103], can be reduced or blocked by ascorbyl compounds [104], as would be predicted by the mutual complementarity of both ascorbate and opioids for amphetamine.

Since the binding of adrenergic compounds (Table 1 and Table 2) to some of the cysteine-containing aminergic receptor peptides (Figure 1 and Table 1 and Table 2) is on a par with their binding to glutathione, it can again be predicted that similar adducts would be formed under conditions of chronic stress or exposure to pharmacological concentrations of adrenochrome compounds. Higher-affinity binding of aminergic compounds to some of the cysteine-containing opioid receptor peptides (Table 1) suggests that these peptide regions may be more susceptible to such adduct formation in the presence of adrenochrome compounds than would be the aminergic receptors—another testable prediction. Notably, glutathione transferases catalyze the detoxication of the o-quinones aminochrome, dopachrome, adrenochrome and noradrenochrome, derived from dopamine, dopa, adrenaline and noradrenaline respectively, protecting cells from the intracellular effects of these reactive compounds [105], but there appears to be no equivalent system at work extracellularly to protect the receptors studied here. The binding of opioid compounds to some of the aminergic receptor peptides tested above also suggests the possibility of functional interactions. For example, Yang et al. [106] reported that naloxone is capable of blocking norepinephrine-induced antinociception. Such an observation makes sense only in a model in which naloxone can interact directly with adrenergic receptors, as has been demonstrated here. Such an aminergic-receptor-mediated mechanism is also consistent with enhancement of adrenergic receptor activity by naloxone and naltrexone [57,58,59,60,61,62,63,64,65].

Adduct formation may also be a consequence of opioid binding to glutathione-like regions of aminergic receptors. While adduct formation between glutathione and opioids has mainly been found to be carried out enzymatically in the liver (see Introduction), Misra and Woods [107] found that morphine can bind directly to glutathione in vitro and, in the presence of ferrous sulfate (a common iron supplement and food antioxidant), produce a covalent adduct. Thus, the binding characterized here can potentially have significant physiological consequences not just for liver and other tissues containing relevant enzymes but also throughout the body. Plasma levels of opiate drugs and their antagonists can reach concentrations of 0.5 to 5 µM or 0.1 to 1 mg/L) under therapeutic and drug use conditions [108]. Since some of the aminergic receptor peptides bound opioids with binding constants between 1 and 30 µM, some binding would be expected to most aminergic receptors and, in the case of the beta 2 adrenergic receptor, it is likely that the opioid binding region would be saturated at these opioid concentrations. Not only would such binding maximize receptor enhancement but, in the presence of any free iron, create the conditions for adduct formation, potentially dysregulating receptor function.

The observation that enkephalins bind to glutathione and glutathione-like peptides may also have important functional implications. While enkephalins are present free in plasma only at 30–60 pM or 60 ng/L [109], their functional concentration in neuronal synaptosomes has been measured to be approximately 2 µM [110] and the intrasynaptic concentration of other neurotransmitters has been calculated to range from a minimum of 2 µM to a maximum of 1.5 mM [111,112,113]. At the binding constants measured in Table 1 and Table 2 above, these enkephalin concentrations translate to anything between one-fifth saturation to complete saturation of the binding sites on the aminergic receptors tested. Such binding is once again theoretically sufficient, given enough time and the presence free iron, to result in the formation of adducts with the glutathione-like sequences of the receptors present in synapses. The same reaction will be catalyzed more efficiently in the liver as discussed in the Introduction.

Binding, without adduct formation, may also modify enkephalin activity within synapses. Ogita et al. [114] have demonstrated that 100 µM met- or leu-enkephalin is able to profoundly diminish glutathione binding to various tissues in a naloxone-independent fashion. Presumably, the mechanism of this antagonism is direct binding of the enkephalins to glutathione producing an inactive complex. Conversely, addition of glutathione to opioid preparations (including enkephalins) antagonizes mu opioid receptor activation [115]. Moreover, glutathione binds with high affinity to N-methyl-D-aspartic acid (NMDA) receptors [116], suggesting a novel mode of NMDA regulation by enkephalins and other opioids using glutathione as an intermediary.

Binding of opioid drugs to adrenergic receptors has, of course, been observed previously, as noted in the Introduction, but the site of the binding has not previously been characterized or associated with glutathione-like regions of aminergic receptors generally. Generalized binding of opioids to non-opioid receptors was observed as soon as attempts were made to identify the first opioid receptors, creating significant technical obstacles to their isolation [117]. For example, Benyhe et al. [118] documented Met5-enkephalin-Arg 6-Phe7 binding to non-opioid receptors as did Webster et al. [119] for benzomorphan. Munro et al. [120] identified one of these non-opioid receptor targets as being adrenergic receptors, observing that three kappa opioid receptor agonists bound in the mid- to high-nanomolar range to alpha-1a adrenoceptors, one acting as a potent functional enhancer. The data generated by this study verify that the enhancer site is the same as has previously been characterized for ascorbic acid enhancement of adrenergic receptors [67,68] and involves glutathione-like sequences. Thus, this study indicates that a mechanistic basis exists for the networking of opioid function and ascorbic acid/antioxidant function.

The implications of these findings for anesthesia, analgesia and substance abuse should be obvious and has recently been reviewed [121]. Any chronic use of opioids and/or adrenergic drugs is likely to deplete glutathione and impair opioid and aminergic receptor function. Fortunately, as was pointed out in the Introduction for opioids, supplemental glutathione or other cysteinyl compounds is able to prevent the deleterious effects of these drugs and are likely to be able to prevent the deleterious effects of adrenergic drugs as well. Ascorbic acid is another potentially valuable compound that may protect both glutathione and the glutathione-like regions of opioid and adrenergic receptors [122]. Indeed, oddly enough, ascorbic acid has been found to have antinociceptive effects at high doses [123,124,125], to be an opiate-sparing adjunctive addition to opioid analgesics [126,127,128], and to moderate some of the effects of withdrawal in opioid addicts [129,130,131]. These effects make sense from the perspective of ascorbic acid and opioids mimicking each other’s binding patterns to glutathione and glutathione-like regions of opioid and aminergic receptors. The heterodimerization of these receptors [78,79] links their activity and the fact that ascorbic acid can bind to both creates a further mechanism for mediating their crosstalk [121].

Whether the impact of anesthesia followed by pain management has the same effects on the antioxidant system as abuse of opioids, amphetamines and other drugs affecting these systems is an issue that needs further research since such effects may be assumed to be a function of drug dosage, means of delivery, length of use and the degree to which tolerance develops. In general, it may be assumed that the use of such drugs for anesthetic purposes will have fewer effects on the antioxidant system than their chronic abuse. For example, typical opioid dosages for anesthesia result in blood plasma concentrations ranging from approximately 1 to 100 nM in blood plasma following oral or intravenous delivery and are typically kept at these levels for only a few days or weeks [132,133]. However, intrathecal delivery of morphine results in transient cerebrospinal fluid concentrations of approximately 25 µM [134], which might result in significant antioxidant effects and covalent reactions despite the brief tissue exposure. Surprisingly, occasional abusers of oral morphine achieve blood concentrations not dissimilar to anesthetized patients, ranging from approximately 6 to 30 nM in blood plasma [135], though their exposure to the drug may last longer and may produce more significant effects on their antioxidant function. Treatment-resistant, chronic opiate abusers are characterized by much higher plasma concentrations ranging from 120 nM to 1 µM [136] and may be presumed to have correspondingly more serious antioxidant deficits. Opioid abusers who must undergo surgery need to achieve concentrations of opioids in the same range as treatment-resistant, chronic abusers [137], underscoring the fact that patients with different drug histories may have different physiological responses to identical drug dosages.

In sum, the antioxidant system, consisting of both intracellular glutathione and extracellular glutathione-like opioid and aminergic receptor regions, plays a fundamental role in mediating opioid and adrenergic drug activity. Ascorbic acid (vitamin C), through its binding to and recycling by glutathione and glutathione-like receptor regions on GPCR, plays a mediating role in all aspects of that opioid and aminergic activity.

## 4. Materials and Methods

### 4.1. Ligands

Opioids (met-enkephalin, morphine sulfate, methadone, and naloxone) and various neurotransmitter and hormonal controls (serotonin, melatonin, histamine, acetylcholine), and adrenergic compounds (epinephrine HCl, norepinephrine HCl, dopamine, L-3,4-dihydroxyphenylalanine (L-DOPA), propranolol, phenylephrine, tyrosine, phenylalanine, and amphetamine), ascorbic acid, dehydroascorbic acid and glucose were obtained from Sigma-Aldrich (St. Louis, MO, USA). Reduced glutathione and oxidized glutathione were obtained from the same source.

### 4.2. Opioid Receptor Peptide Synthesis and Preparation

The peptides listed in Figure 1 were synthesized to at least 95% purity (as determined by mass spectrometry) by RS Synthesis (Louisville, KY, USA). Each of these receptor peptides were made into individual stock solutions with a concentration of 1 mg/mL in pH 7.00 phosphate-buffered saline solution (Fischer Scientific, Hampton, NH, USA). For each individual test, the concentration of the stock solution was calculated and then diluted to a 10 µM solution. Each of these peptide solutions was tested for binding with the various opioid, adrenergic and ascorbate-like compounds and controls such as histamine and acetylcholine (Sigma-Aldrich, St. Louis, MO, USA), as indicated in Table 1 and Table 2. The opioid, adrenergic, ascorbyl and control compounds were made up at 1.0 mM solutions and then diluted to 5 µM concentrations for use in the peptide binding experiments.

### 4.3. Peptide Binding Test Methods

After the solutions were made, a 96-well quartz crystal plate was prepared to be run through the spectrophotometer at room temperature (ca. 24 °C). The plate was set up to have the absorbance of each of the adrenergic compound dilutions measured on their own, and with each receptor peptide. The absorbance of each receptor peptide without the presence of the adrenergic compound was measured as well. The absorbance of each well was measured at every 10 nm increment from 190–260 nm. The maximum absorbance that can be measured was set to be 4. Each well had 200 μL of solution, so if the absorbance of one component was being measured, it was diluted by ½ with phosphate buffer. Thus, the final concentration of receptor peptide in each well varied from 0.5 mM to 0.05 pM while the other compound was held fixed at 5 µM.

Spectrophotometry (SPECTRAmax plus scanning spectrophotometer with the SOFTmax PRO program) was used to measure the binding between opioid receptor peptides and opioid, adrenergic and control compounds. The method of calculating binding constants was to measure the absorbance (minus the absorbance of the buffer) of 100 µL of each compound by itself diluted by 100 µL of buffer at a given wavelength (in this case at 200 nm) at the range of concentrations utilized in the experiments. The average of the resulting values is calculated. (If the experiments are run properly, there is very little variance—a couple of percent. If there was more variance than that, the experiment was re-run.) An “expected” value is calculated following Beer’s law, by adding the absorbances of each pair of compounds (peptide plus small molecule). The actual chemical combinations of these molecules was then run under identical conditions.

### 4.4. Data Analysis

The absorbance for the phosphate buffer is subtracted for each well before any calculations are performed. The difference between the “expected” and actual values was determined at each concentration of the varied compound. This difference was then plotted against the concentration of the varied compound. If there was significant binding, an S-shaped curve resulted. The binding constant was approximated by using the inflection point of the binding curve (i.e., the point at which the dissociation and association of the two molecules was equal). If there was no significant binding, no curve resulted or the curve did not inflect over the range of concentrations utilized in the experiments, in which case the binding constant was greater than could be calculated from the experimental conditions. All data were analyzed using Microsoft Excel.

As noted above, all combinations were in duplicate and every combination was repeated in independent experiments at least twice, yielding the same Kd. Because the resulting absorbances are the differences between the independently measured absorbances, it is not possible to calculate error bars or to perform statistics on the resulting values.

## Figures and Tables

**Figure 1 ijms-21-06230-f001:**
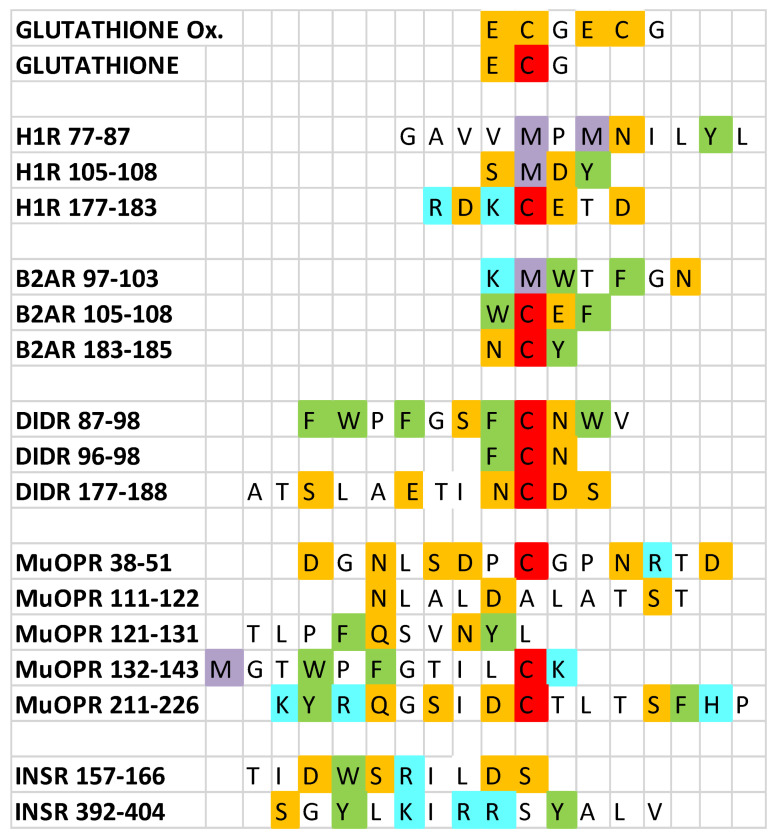
Sequences of the peptides utilized in the experiments reported here. Ox = oxidized; H1R = histamine 1 receptor; B2AR = beta 2 adrenergic receptor; D1DR = dopamine 1 receptor; muOPR = mu opioid receptor; INSR = insulin receptor. Amino acids are indicated by their one-letter abbreviations and are colored according to their charge similarities: blue = positively charged; orange = negatively charged; green = aromatic; white = neutral; red = cysteine; purple = methionine. Note that the cysteines in oxidized glutathione are colored orange because they are involved in a disulfide bond rather than being free cysteines (red).

**Figure 2 ijms-21-06230-f002:**
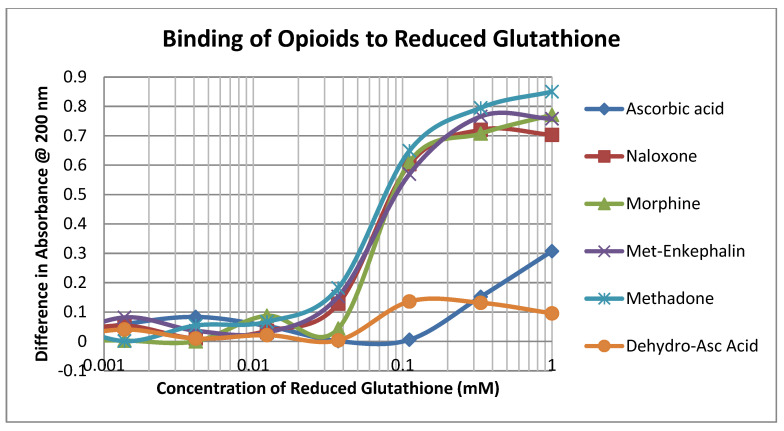
Binding of opioid drugs, ascorbic acid, and dehydroascorbic acid to reduced glutathione.

**Figure 3 ijms-21-06230-f003:**
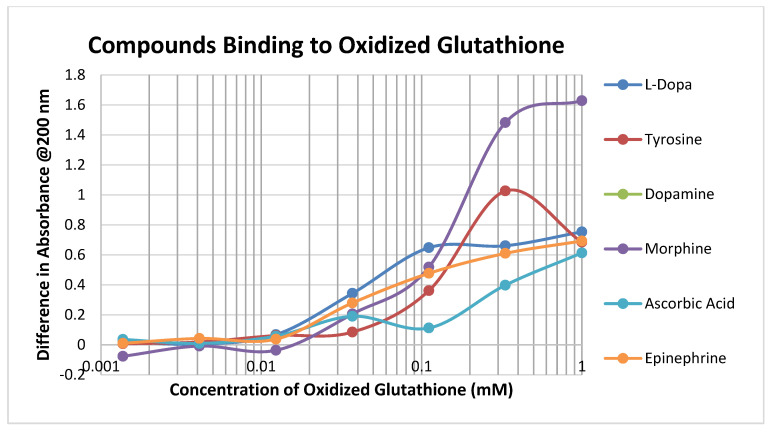
Binding of diverse adrenergic and opioid compounds and ascorbic acid to oxidized glutathione. Comparison with Figure 2 demonstrates that binding affinities are generally lower to oxidized glutathione than to reduced glutathione (see Table 1 and Table 2 for additional comparisons).

**Figure 4 ijms-21-06230-f004:**
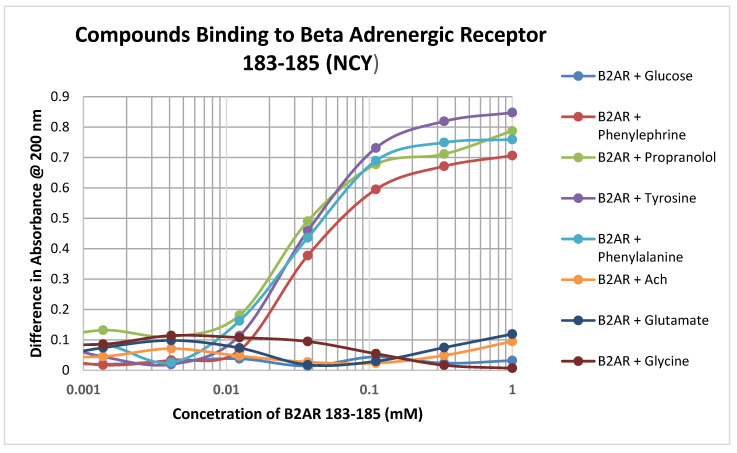
Binding of various adrenergic compounds and controls to the glutathione-like peptide NCY derived from the beta 2 adrenergic receptor. See Table 2 for additional data.

**Figure 5 ijms-21-06230-f005:**
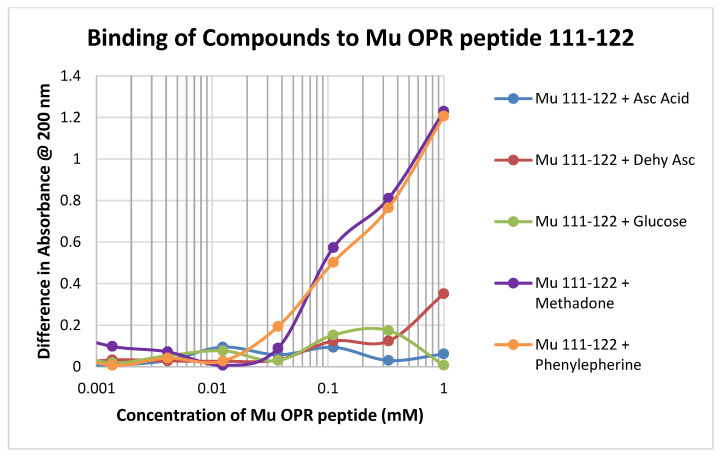
Binding of various compounds to the muOPR peptide 111–122. The observed lack of significant binding was typical of all of the peptides lacking cysteine residues (see Figure 1 and Table 1).

**Figure 6 ijms-21-06230-f006:**
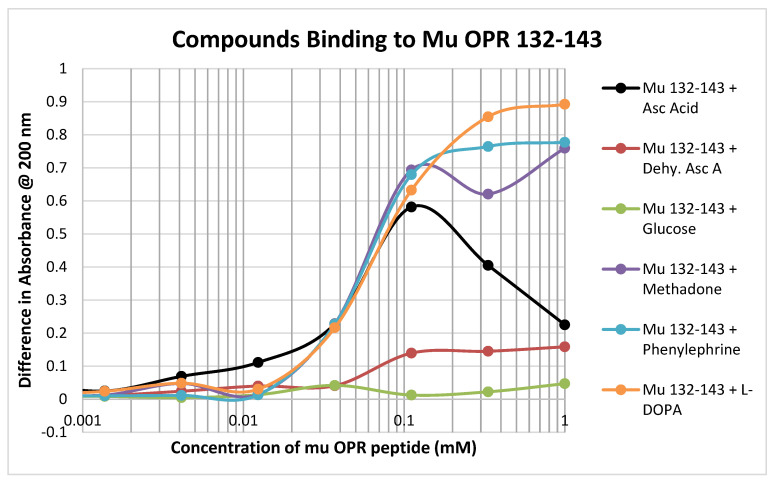
Binding of various compounds to muOPR 132–143. Compare with Figure 5. Additional data in Table 1.

**Figure 7 ijms-21-06230-f007:**
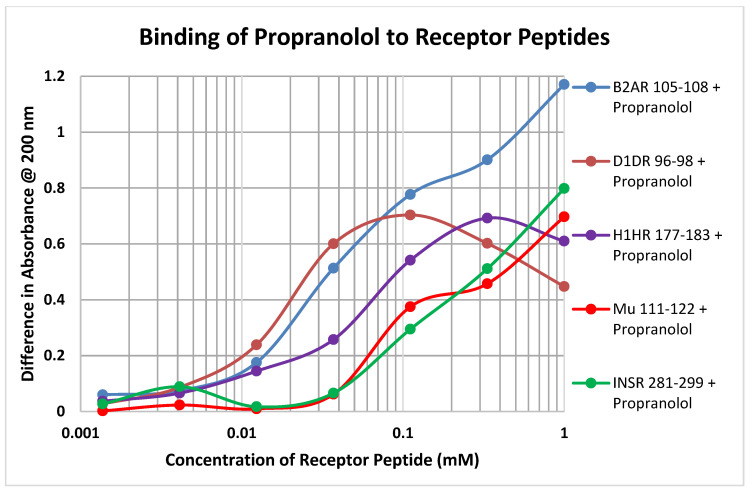
Binding of propranolol to receptor peptide derived from the beta 2 adrenergic receptor (B2AR), dopamine 1 receptor (D1DR), histamine 1 receptor (H1HR), mu opioid receptor (Mu) and insulin receptor (INSR). Additional data in Table 1 and Table 2.

**Table 1 ijms-21-06230-t001:** Summary of ultraviolet spectroscopy studies of the binding of various compounds to peptides: Glut RED = reduced glutathione; Glut Ox = oxidized glutathione; Mu OPR = mu opioid receptor; INSR = insulin receptor; Dehydro Asc = dehydroascorbic acid; L-DOPA = L-3,4-dihydroxyphenylalanine.

Kd (µM)	Glut RED ECG	Glut O_X_ ECG-ECG	Mu OPR 38–51	Mu OPR 111–122	Mu OPR 121–131	Mu OPR 132–143	Mu OPR 211–226	INSR 157–166	INSR 392–404
Ascorbic Acid	60	20/310	70	>1000	>1000	5/40	65/700	100	>1000
Dehydro Asc	60	60	>1000	>1000	>1000	>1000	150	150	>1000
Glucose	>1000	>1000	>1000	>1000	>1000	>1000	>1000	>1000	>1000
Morphine	60	300	35	50	900	35	30	60	>1000
Methadone	60	300	70	70	>1000	50	150	80	>1000
Naloxone	50	300	0.5/35	0.5/38	>1000	0.5/42	1.0/45	200	>1000
Met-Enkephalin	70	400	1.2/35	0.33/80	3.5/90	0.4/70	1.0/65	100	>1000
Phenylephrine	5	12	1.0/30	150	>1000	60	150	70	>1000
Propranolol	5	9	25	70	>1000	90	200	300	700
Amphetamine	20	60	1.3/90	1.3/100	>1000	1.1/85	1.2/90	400	>1000
Epinephrine	40	63	1.2/35	1.3/40	>1000	1.4/35	1.2/45	>1000	200
Norepinephrine	110	60	1.4/45	1.3/40	>1000	1.4/40	1.3/45	140	>1000
Dopamine	90	90	60	65	>1000	60	65	400	>1000
L-DOPA	50	50	80	60	>1000	70	150	90	>1000
Tyrosine	130	130	85	700	>1000	60	160	90	>1000
Phenylalanine	>1000	90	50	>1000	>1000	80	200	90	>1000
Histamine	500	>1000	>1000	>1000	>1000	>1000	>1000	110	>1000
Serotonin	70	>1000	100	100	350	100	90	110	900
Melatonin	100	>1000	50	400	>1000	65	130	80	>1000
Acetylcholine	>1000	>1000	>1000	>1000	>1000	>1000	>1000	>1000	>1000
Glutamate	>1000	>1000	>1000	>1000	>1000	>1000	>1000	>1000	>1000
Glycine	>1000	>1000	>1000	>1000	>1000	>1000	>1000	>1000	>1000

**Table 2 ijms-21-06230-t002:** Summary of ultraviolet spectroscopy studies of the binding of various compounds to peptides: Glut RED = reduced glutathione; Glut Ox = oxidized glutathione; B2AR = beta 2 adrenergic receptor; D1DR = dopamine 1 receptor; H1HR = histamine 1 receptor; Dehydro Asc = dehydroascorbic acid; L-DOPA = L-3,4-dihydroxyphenylalanine.

Kd (µM)	B2AR 97–103 KMWTFGN	B2AR 105–108 WCEF	B2AR 183–185 NCY	D1DR 89–100 FWPFGSFCN	D1DR 96–98 FCN	DRD1 177–188 ATSLAETINCDS	H1HR 77–87 GAVVMPMNILYL	H1HR 105–108 SMDY	HIHR 177–183 RDKCETD
Ascorbic Acid	65	35	12	>1000	7	300	300	60	130
Dehydro Asc	60	20	500	>1000	20	200	35	15	>1000
Glucose	>1000	>1000	>1000	>1000	>1000	>1000	60	>1000	>1000
Morphine	1	30	>1000	30	25	310	110	50	30
Methadone	300	30	>1000	120	25	110	50	10	150
Naloxone	6	30	>1000	60	30	150	110	50	40
Met-Enkephalin	130	30	>1000	10	30	150	2.3/70	3.0/70	55
Phenylephrine	>1000	35	30	>1000	50	80	>1000	50	50
Propranolol	>1000	40	22	>1000	20	80	>1000	45	50
Amphetamine	130	20	2.3	530	2.5	230	60	70	35
Epinephrine	120	35	25	400	40	900	30	60	40
Norepinephrine	600	35	12	300	50	1000	30	30	45
Dopamine	>1000	30	35	750	50	300	30	35	50
L-DOPA	200	40	25	>1000	60	210	90	35	55
Tyrosine	>1000	50	35	>1000	22	80	>1000	50	50
Phenylalanine	>1000	55	30	>1000	25	130	>1000	60	50
Histamine	>1000	50	>1000	600	50	>1000	20	70	70
Serotonin	430	45	>1000	>1000	30	>1000	50	120	60
Melatonin	600	25	>1000	750	22	130	55	50	55
Acetylcholine	>1000	>1000	>1000	>1000	>1000	>1000	>1000	>1000	>1000
Glutamate	>1000	>1000	>1000	>1000	>1000	>1000	>1000	>1000	>1000
Glycine	>1000	>1000	>1000	>1000	>1000	>1000	>1000	>1000	>1000

**Table 3 ijms-21-06230-t003:** Overall patterns of binding between classes of compounds and receptor types. ^ Kd < or = 60 µM but >15 µM/Kd < or = 15 µM. Classes of compounds that bound to a receptor type at both cutoff values are shown bolded on a medium gray background; classes of compounds that bound to a receptor type only between 16 and 60 µM are bolded on a light gray background, while those that met neither cutoff value are normal type on a white background.

BINDING ^	Ascorbate	Opioids	Adrenergics	Serotonin Melatonin	Histamine	Acetyl-Choline	Glutamate	Glucose
Glutathione	**YES/no**	**YES/no**	**YES/YES**	no/no	no/no	no/no	no/no	no/no
Opioid Receptor	**YES/YES**	**YES/YES**	**YES/YES**	no/no	no/no	no/no	no/no	no/no
Adrenergic Receptor	**YES/YES**	**YES/YES**	**YES/YES**	**YES/no**	**YES/no**	no/no	no/no	no/no
Dopamine Receptor	**YES/YES**	**YES/no**	**YES/YES**	**YES/no**	**YES/no**	no/no	no/no	no/no
Histamine Receptor	**YES/no**	**YES/YES**	**YES/no**	**YES/no**	**YES/YES**	no/no	no/no	no/no
Insulin Receptor	no/no	no/no	no/no	no/no	no/no	no/no	no/no	no/no

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
