# Peer review of "Glutathione and Glutathione-Like Sequences of Opioid and Aminergic Receptors Bind Ascorbic Acid, Adrenergic and Opioid Drugs Mediating Antioxidant Function: Relevance for Anesthesia and Abuse"

_ijms, 2020, doi:10.3390/ijms21176230_

Round 1
Reviewer 1 Report
The authors have appropriately revised the manuscript, which is now suitable for publications.
Author Response
Thank you for your suggestions!
Reviewer 2 Report
Thanks for the detailed responses. Two minor concerns remain.
- It still doesn’t make sense to me for the replication. Should each replication experiment yield a Kd value? If repeated, would it have two independent Kd values? Were the Kd values listed on Table 2 the means of two independent experiments?
- Based on Methods, receptor peptides were used at a fixed 5 uM concentration, while compounds were from 0.05 pM to 0.5 mM. However, all x-axis of the figures was labeled as Concentration of peptides arranging 0.001 – 1 mM. Was the x-axis mislabeled? If not, my understanding was that you were doing a binding study using various concentrations of receptor peptides as “ligand” against a fixed concentration of compounds and that’s why my question was raised. If the experiment was designed in other way around as described in Methods, the x-axis should be labeled as compound concentration. Or you need to have a better description. Otherwise, it would be confused for an outside reader.
Author Response
1) Yes, each replication should yield a Kd value and should yield the same Kd value. The actual shape of the curve may vary slightly from run to run, but the inflection point should not. If it does, then the dilutions were not made properly and we re-ran the combination until a stable Kd emerged. (This was a very rare event!)
2) My mistake in rewriting the Methods. The peptides were subjected to serial dilution and the other compound held constant at 5 uM. This is now corrected.
Reviewer 3 Report
Summary:
In this paper, the authors investigated the relative binding affinities of ascorbic acid, dehydroascorbic acid, opioid and adrenergic compounds to glutathione and glutathione-like peptides derived from the extracellular loop regions of the human beta-2-adrenergic, dopamine D1, histamine H1, and mu-opioid receptors. This work was based on presumed functional interactions among opioids and amines, their receptors, and glutathione shown in previous studies on effects of these compounds on antioxidant functions related to anesthesia and addiction.
Broad comments:
The authors aim to gain more information on opioid and adrenergic drug activity. However, binding of peptide fragments derived from GPCR does not provide sufficient information about functions and interactions of intact GPCR. Additional assays (e.g. for G-protein activation, G-protein dissociation) have to be applied. This greatly limits the significance and overall merit of this paper. No measures to reduce bias (e.g. blinding) and no statistical methods for data analysis were used. This is not acceptable.
Specific comments:
The introduction is too long and not focused on the study at hand. It should be shortened by about two thirds. Several passages can be moved to the discussion. The aim of study is explained in the first paragraph in the introduction without references. Even though publications are cited later in the manuscript, they should be cited when they are first mentioned. It would be easier for the reader to understand the purpose of the study if the first paragraph was moved to the end of the introduction. On lines 52-53 the authors indicate that opioids commonly cause liver and kidney damage and cite several references (# 3-10). The authors should explicitly state how such effects were demonstrated, e.g. whether these were in vitro or in vivo effects, in which species, at which concentrations etc.. The reader must be able to judge whether such previously published reports are relevant to the clinical situation in humans. The same applies to the statements on lines 105 and 115.
The authors indicate that negative effects of opioid and adrenergic compounds on anti-oxidation functions related to anesthesia and addiction could be due to their binding to glutathione and glutathione-like regions of their receptors. However, the amounts of opioids used by humans misusing opioids (the term “addiction” includes psychological aspects and should not be used here) versus during anesthesia are very different. Thus, opioid metabolism and interactions with glutathione and glutathione-like regions of opioid and adrenergic receptors would differ. These differences must be outlined in the discussion.
The authors state on lines 45-46 that the mechanisms of interactions between opioid and adrenergic receptors are not fully explored. Unfortunately, the present study investigates only binding to certain fragments of receptors. Binding is only one out of several important components of CPCR function (e.g. changes of GPCR conformation, G-protein coupling, G-protein dissociation, downstream signaling pathways). Binding of short fragments of receptors does not provide information about binding of the intact receptor or about the other components involved in functions or interactions between different receptors. This greatly limits the contribution of this study to clarify the initial questions posed, i.e. the interactions mentioned at the outset. This must be clearly stated in the discussion. The sensitivity of ultraviolet spectrophotometry technique used in this study is solid and has been validated with other techniques. However, the use of only one technique is very limited. It would be better to confirm promising peptide and drug interactions with other methods as well.
Apparently, no measures to preclude bias (blinding) and no methods of statistical analysis were used. This is not acceptable. After determination of sample sizes (power analysis), the experiments should be repeated several times in a blinded manner and the data must be analyzed with the help of a statistician. Only statistically significant results should be discussed. It is misleading to discuss tendencies. These passages should be omitted from the paper.
How did the authors determine the length and the characteristics of peptide fragments? There is no clear explanation.
Although Kd values listed for the binding of opioids and ascorbic acid and dehydro-ascorbic acid in table 2 are close to each other (around 60 uM), the graphical illustration in figure 1 between opioids and ascorbic acid & dehydro-ascorbic acid looks very different. How do the authors explain this?
The figure legends should be more detailed.
Please write the name of the peptides fragments when stating their affinity to ascorbic acid on lines 261 and 262.
Please correct the statement on line 269 about affinity of ascorbic acid and dehydroascorbic acid to aminergic receptors ranging from 5-65 uM where kd is starting smallest from 7.
There is no figure 7 in the publication. Please correct your statement on line 289.
Ad line 465 and 469: The relationship between the laboratory and the sponsor (Maurine Bernstein) should be explained.
Author Response
Summary:
In this paper, the authors investigated the relative binding affinities of ascorbic acid, dehydroascorbic acid, opioid and adrenergic compounds to glutathione and glutathione-like peptides derived from the extracellular loop regions of the human beta-2-adrenergic, dopamine D1, histamine H1, and mu-opioid receptors. This work was based on presumed functional interactions among opioids and amines, their receptors, and glutathione shown in previous studies on effects of these compounds on antioxidant functions related to anesthesia and addiction.
Broad comments:
The authors aim to gain more information on opioid and adrenergic drug activity. However, binding of peptide fragments derived from GPCR does not provide sufficient information about functions and interactions of intact GPCR. Additional assays (e.g. for G-protein activation, G-protein dissociation) have to be applied. This greatly limits the significance and overall merit of this paper. No measures to reduce bias (e.g. blinding) and no statistical methods for data analysis were used. This is not acceptable.
Please read the TITLE of the article: the research is NOT about GPCR function; it is about whether binding of drugs to glutathione-like regions of GPCR may modify their antioxidant function. The Introduction reviews dozens of studies that have ALREADY demonstrated that the sets of compounds tested here have GPCR functions – what this study does that is new is to test whether these compounds bind SPECIFICALLY TO THE GLUTATIONE-LIKE REGIONS OF THE RECEPTORS. There is ABSOLUTELY NO NEED to demonstrate functionality of these compounds because there are literally thousands of papers that have already done so. So, yes, the narrow focus limits the significance and overall merit of the paper – we aren’t claiming to have discovered new classes of drugs that modify GPCR functions – however, we do claim to be the first to characterize whether these compounds bind to specific, glutathione-like regions of GPCR (which, by the way, we have already demonstrated to be able to carry out glutathione-like functions as well). Doing the functional studies requested by the Reviewer would be a complete waste of time because they have already been done.
The issue of “experimenter bias” is ludicrous. Blinding of “treatments” is only used when one of two conditions are met. One is when the subject of the experiment can be influenced by knowing what their treatment is; the other is when the experimenter must use judgement in the role of observer/recorder of the outcome of an experiment and might unintentionally modify the data to fit their preconceptions. In this case, the subjects of the experiments are chemicals that cannot be influenced by “knowing” what their partner is. Neither can bias be introduced by the experimenter in this case because no judgement is involved: both the observation and recording of the data are performed by a machine. There is no place in the experiments where the experimenter can inadvertently modify the results. Thus, there is no need or purpose in “blinding” the experiments. Moreover, we note that we have never read ANY chemistry paper (which is what ours is) in which such blinding was carried out or required and we consider this demand to be extraordinary and irrelevant.
The Methods section already addresses why it is impossible to apply statistical methods to the data analysis: binding curves are generated (following the standard procedure in the field) by adding the spectra of the individual compounds to each other and taking the difference from the spectrum of the actual combination. No statistical method exists for carrying over error bars or standard deviations or standard errors through these additions and subtractions. The Reviewer is therefore asking us to do something that is literally impossible. Whether the Reviewer finds that acceptable or not, that is the plain truth of the matter. Not only can we not provide such statistics; no one would be able to do so.
Specific comments:
The introduction is too long and not focused on the study at hand. It should be shortened by about two thirds. Several passages can be moved to the discussion. The aim of study is explained in the first paragraph in the introduction without references. Even though publications are cited later in the manuscript, they should be cited when they are first mentioned. It would be easier for the reader to understand the purpose of the study if the first paragraph was moved to the end of the introduction.
This is obviously a stylistic matter. We disagree with the Reviewer and note that the other two Reviewers had no problems with the Introduction as written. Moreover, if we did as the Reviewr suggests, we would then leave the reader with no idea what the study is about until the very end of the Introduction. That makes no sense. We choose to leave it as it is.
On lines 52-53 the authors indicate that opioids commonly cause liver and kidney damage and cite several references (# 3-10). The authors should explicitly state how such effects were demonstrated, e.g. whether these were in vitro or in vivo effects, in which species, at which concentrations etc.. The reader must be able to judge whether such previously published reports are relevant to the clinical situation in humans. The same applies to the statements on lines 105 and 115.
Done.
This issue of whether the concentrations of drugs utilized in the various studies is, in any event, addressed at much greater length in the Discussion section from lines 300 on.
The authors indicate that negative effects of opioid and adrenergic compounds on anti-oxidation functions related to anesthesia and addiction could be due to their binding to glutathione and glutathione-like regions of their receptors. However, the amounts of opioids used by humans misusing opioids (the term “addiction” includes psychological aspects and should not be used here) versus during anesthesia are very different. Thus, opioid metabolism and interactions with glutathione and glutathione-like regions of opioid and adrenergic receptors would differ. These differences must be outlined in the discussion.
“Addiction” replaced with “abuse” throughout. New paragraph addressing this issue added at line 405 accompanied by new references 132-137.
The authors state on lines 45-46 that the mechanisms of interactions between opioid and adrenergic receptors are not fully explored. Unfortunately, the present study investigates only binding to certain fragments of receptors. Binding is only one out of several important components of CPCR function (e.g. changes of GPCR conformation, G-protein coupling, G-protein dissociation, downstream signaling pathways). Binding of short fragments of receptors does not provide information about binding of the intact receptor or about the other components involved in functions or interactions between different receptors. This greatly limits the contribution of this study to clarify the initial questions posed, i.e. the interactions mentioned at the outset. This must be clearly stated in the discussion.
Added at line 245 (intro to Discussion).
The sensitivity of ultraviolet spectrophotometry technique used in this study is solid and has been validated with other techniques. However, the use of only one technique is very limited. It would be better to confirm promising peptide and drug interactions with other methods as well.
We’ve been doing these studies for 25 years and have validated the technique with other methods multiple times (in papers that are cited in the Methods) as have other people (also cited in the Methods). We are not going to waste our time revisiting a question that has been settled multiple times decades ago by multiple labs. Indeed, no one who knows the history of chemistry should be requesting such validation in light of the fact that Wilhelm Ostwald won the Nobel Prize in 1909 for demonstrating extensively, both theoretically and experimentally, that all colligative properties are interchangeable so that it does not matter which one is used for measurement.
Apparently, no measures to preclude bias (blinding) and no methods of statistical analysis were used. This is not acceptable. After determination of sample sizes (power analysis), the experiments should be repeated several times in a blinded manner and the data must be analyzed with the help of a statistician. Only statistically significant results should be discussed. It is misleading to discuss tendencies. These passages should be omitted from the paper.
This is nonsensical and completely inappropriate to a chemistry paper, as discussed above.
How did the authors determine the length and the characteristics of peptide fragments? There is no clear explanation.
Yes, we did: they were synthesized. That means their composition is known. In fact, their composition and purity were determined by mass spectrometry by the manufacturer (as was already stated in the Methods), who provided the data to us with the peptides and the mass spectrometry data, as is standard practice in the industry.
Although Kd values listed for the binding of opioids and ascorbic acid and dehydro-ascorbic acid in table 2 are close to each other (around 60 uM), the graphical illustration in figure 1 between opioids and ascorbic acid & dehydro-ascorbic acid looks very different. How do the authors explain this?
Binding constants do not depend on the height of the curve, only one the inflection point. The height of the curve is a function of the absorbance; ascorbic acid has a robust absorbance at 200 nm while dehydroascorbate has very little absorbance at 200 nm and therefore when one does the additions and subtractions to produce the binding curve, the dehydroascorbate-opioid curves are much smaller. The size of the curve has no relationship to whether there is a significant difference in expected and observed values.
The figure legends should be more detailed.
This was not a problem for the other two Reviewers and is too vague to address in any meaningful way. Our goal in the captions was not to repeat information that is already in the text and our style is to allow readers to interpret the data for themselves.
Please write the name of the peptides fragments when stating their affinity to ascorbic acid on lines 261 and 262.
Done
Please correct the statement on line 269 about affinity of ascorbic acid and dehydroascorbic acid to aminergic receptors ranging from 5-65 uM where kd is starting smallest from 7.
Done
There is no figure 7 in the publication. Please correct your statement on line 289.
Done
Ad line 465 and 469: The relationship between the laboratory and the sponsor (Maurine Bernstein) should be explained.
Maurine Bernstein is the mother of Robert Root-Bernstein. How is this possibly relevant?
Round 2
Reviewer 3 Report
In this paper, the authors investigated the relative binding affinities of ascorbic acid, dehydroascorbic acid, opioid and adrenergic compounds to glutathione and glutathione-like peptides derived from the extracellular loop regions of the human beta-2-adrenergic, dopamine D1, histamine H1, and mu-opioid receptors. This work was based on presumed functional interactions among opioids and amines, their receptors, and glutathione shown in previous studies on effects of these compounds on antioxidant functions related to anesthesia and addiction.
Broad comments:
The authors aim to gain more information on opioid and adrenergic drug activity. However, binding of peptide fragments derived from GPCR does not provide sufficient information about functions and interactions of intact GPCR. Additional assays (e.g. for G-protein activation, G-protein dissociation) have to be applied. This greatly limits the significance and overall merit of this paper. No measures to reduce bias (e.g. blinding) and no statistical methods for data analysis were used. This is not acceptable.
- Authors: Please read the TITLE of the article: the research is NOT about GPCR function; it is about whether binding of drugs to glutathione-like regions of GPCR may modify their antioxidant function. The Introduction reviews dozens of studies that have ALREADY demonstrated that the sets of compounds tested here have GPCR functions – what this study does that is new is to test whether these compounds bind SPECIFICALLY TO THE GLUTATIONE-LIKE REGIONS OF THE RECEPTORS. There is ABSOLUTELY NO NEED to demonstrate functionality of these compounds because there are literally thousands of papers that have already done so. So, yes, the narrow focus limits the significance and overall merit of the paper – we aren’t claiming to have discovered new classes of drugs that modify GPCR functions – however, we do claim to be the first to characterize whether these compounds bind to specific, glutathione-like regions of GPCR (which, by the way, we have already demonstrated to be able to carry out glutathione-like functions as well). Doing the functional studies requested by the Reviewer would be a complete waste of time because they have already been done.
- Response:
The title is misleading and the introduction is too long. Anti-oxidant “function” was not measured in this paper, neither was the “relevance for anesthesia and abuse”. The issues mentioned by the authors need to be made clear to the reader by altering the title, and by shortening and focusing the introduction.
- Authors: The issue of “experimenter bias” is ludicrous. Blinding of “treatments” is only used when one of two conditions are met. One is when the subject of the experiment can be influenced by knowing what their treatment is; the other is when the experimenter must use judgement in the role of observer/recorder of the outcome of an experiment and might unintentionally modify the data to fit their preconceptions. In this case, the subjects of the experiments are chemicals that cannot be influenced by “knowing” what their partner is. Neither can bias be introduced by the experimenter in this case because no judgement is involved: both the observation and recording of the data are performed by a machine. There is no place in the experiments where the experimenter can inadvertently modify the results. Thus, there is no need or purpose in “blinding” the experiments. Moreover, we note that we have never read ANY chemistry paper (which is what ours is) in which such blinding was carried out or required and we consider this demand to be extraordinary and irrelevant.
- Response:
Agreed, these problems occur throughout the chemical literature and have significantly contributed to its devaluation. Blinding is required in all scientific disciplines, as has been discussed repeatedly by many authors (see e.g. Nature 2020;584:9; Nature 2015;526:187; National Academies of Sciences, Engineering, and Medicine 2019. Reproducibility and Replicability in Science. Washington, DC: The National Academies Press. https://doi.org/10.17226/25303).
- Authors: The Methods section already addresses why it is impossible to apply statistical methods to the data analysis: binding curves are generated (following the standard procedure in the field) by adding the spectra of the individual compounds to each other and taking the difference from the spectrum of the actual combination. No statistical method exists for carrying over error bars or standard deviations or standard errors through these additions and subtractions. The Reviewer is therefore asking us to do something that is literally impossible. Whether the Reviewer finds that acceptable or not, that is the plain truth of the matter. Not only can we not provide such statistics; no one would be able to do so.
- Response:
The comment remains unanswered. Statistical analysis of the data is definitely necessary for publication in a scientific journal (see e.g. Nature 2020;584:9; National Academies of Sciences, Engineering, and Medicine 2019. Reproducibility and Replicability in Science. Washington, DC: The National Academies Press. https://doi.org/10.17226/25303).
Specific comments:
The introduction is too long and not focused on the study at hand. It should be shortened by about two thirds. Several passages can be moved to the discussion. The aim of study is explained in the first paragraph in the introduction without references. Even though publications are cited later in the manuscript, they should be cited when they are first mentioned. It would be easier for the reader to understand the purpose of the study if the first paragraph was moved to the end of the introduction.
- Authors: This is obviously a stylistic matter. We disagree with the Reviewer and note that the other two Reviewers had no problems with the Introduction as written. Moreover, if we did as the Reviewr suggests, we would then leave the reader with no idea what the study is about until the very end of the Introduction. That makes no sense. We choose to leave it as it is.
- Response:
The introduction is now even longer and is still not sufficiently focused. It needs to be rewritten to succinctly familiarize the reader with the purpose and background of the study.
On lines 52-53 the authors indicate that opioids commonly cause liver and kidney damage and cite several references (# 3-10). The authors should explicitly state how such effects were demonstrated, e.g. whether these were in vitro or in vivo effects, in which species, at which concentrations etc.. The reader must be able to judge whether such previously published reports are relevant to the clinical situation in humans. The same applies to the statements on lines 105 and 115.
- Authors: Done.
This issue of whether the concentrations of drugs utilized in the various studies is, in any event, addressed at much greater length in the Discussion section from lines 300 on.
- Response:
Line 108: What are “physiologically relevant concentrations” of morphine, enkephalins and naloxone? Morphine and naloxone are exogenous substances that do not occur physiologically. In which anatomical regions do interactions between enkephalins (Met- or Leu-enkephalin?) and aminergic receptors (which ones?) occur and at which concentrations? Lines 117-118: Similarly, aminergic drugs are exogenous substances that do not occur physiologically.
The authors indicate that negative effects of opioid and adrenergic compounds on anti-oxidation functions related to anesthesia and addiction could be due to their binding to glutathione and glutathione-like regions of their receptors. However, the amounts of opioids used by humans misusing opioids (the term “addiction” includes psychological aspects and should not be used here) versus during anesthesia are very different. Thus, opioid metabolism and interactions with glutathione and glutathione-like regions of opioid and adrenergic receptors would differ. These differences must be outlined in the discussion.
- Authors: “Addiction” replaced with “abuse” throughout. New paragraph addressing this issue added at line 405 accompanied by new references 132-137.
- Response:
Please do not use stigmatizing terms like ‘addicted’ or ‘abuse’. These may be replaced with ‘use’ or ‘non-medical use’ as appropriate.
The authors state on lines 45-46 that the mechanisms of interactions between opioid and adrenergic receptors are not fully explored. Unfortunately, the present study investigates only binding to certain fragments of receptors. Binding is only one out of several important components of CPCR function (e.g. changes of GPCR conformation, G-protein coupling, G-protein dissociation, downstream signaling pathways). Binding of short fragments of receptors does not provide information about binding of the intact receptor or about the other components involved in functions or interactions between different receptors. This greatly limits the contribution of this study to clarify the initial questions posed, i.e. the interactions mentioned at the outset. This must be clearly stated in the discussion.
- Authors: Added at line 245 (intro to Discussion).
- Response: o.k.
The sensitivity of ultraviolet spectrophotometry technique used in this study is solid and has been validated with other techniques. However, the use of only one technique is very limited. It would be better to confirm promising peptide and drug interactions with other methods as well.
- Authors: We’ve been doing these studies for 25 years and have validated the technique with other methods multiple times (in papers that are cited in the Methods) as have other people (also cited in the Methods). We are not going to waste our time revisiting a question that has been settled multiple times decades ago by multiple labs. Indeed, no one who knows the history of chemistry should be requesting such validation in light of the fact that Wilhelm Ostwald won the Nobel Prize in 1909 for demonstrating extensively, both theoretically and experimentally, that all colligative properties are interchangeable so that it does not matter which one is used for measurement.
- Response:
The comment concerned “interactions”, which have not been demonstrated here.
Apparently, no measures to preclude bias (blinding) and no methods of statistical analysis were used. This is not acceptable. After determination of sample sizes (power analysis), the experiments should be repeated several times in a blinded manner and the data must be analyzed with the help of a statistician. Only statistically significant results should be discussed. It is misleading to discuss tendencies. These passages should be omitted from the paper.
- Authors: This is nonsensical and completely inappropriate to a chemistry paper, as discussed above.
- Response:
See above.
How did the authors determine the length and the characteristics of peptide fragments? There is no clear explanation.
- Authors: Yes, we did: they were synthesized. That means their composition is known. In fact, their composition and purity were determined by mass spectrometry by the manufacturer (as was already stated in the Methods), who provided the data to us with the peptides and the mass spectrometry data, as is standard practice in the industry.
- Response:
This question remains unanswered. The reader needs to know why these specific peptide fragments were chosen.
Although Kd values listed for the binding of opioids and ascorbic acid and dehydro-ascorbic acid in table 2 are close to each other (around 60 uM), the graphical illustration in figure 1 between opioids and ascorbic acid & dehydro-ascorbic acid looks very different. How do the authors explain this?
- Authors: Binding constants do not depend on the height of the curve, only one the inflection point. The height of the curve is a function of the absorbance; ascorbic acid has a robust absorbance at 200 nm while dehydroascorbate has very little absorbance at 200 nm and therefore when one does the additions and subtractions to produce the binding curve, the dehydroascorbate-opioid curves are much smaller. The size of the curve has no relationship to whether there is a significant difference in expected and observed values.
- Response: o.k.
The figure legends should be more detailed.
- Authors:This was not a problem for the other two Reviewers and is too vague to address in any meaningful way. Our goal in the captions was not to repeat information that is already in the text and our style is to allow readers to interpret the data for themselves.
- Response: o.k.
Please write the name of the peptide fragments when stating their affinity to ascorbic acid on lines 261 and 262.
- Authors: Done
- Response: o.k.
Please correct the statement on line 269 about affinity of ascorbic acid and dehydroascorbic acid to aminergic receptors ranging from 5-65 uM where kd is starting smallest from 7.
- Authors: Done
- Response: o.k.
There is no figure 7 in the publication. Please correct your statement on line 289.
- Authors: Done
- Response: o.k.
Ad line 465 and 469: The relationship between the laboratory and the sponsor (Maurine Bernstein) should be explained.
- Authors: Maurine Bernstein is the mother of Robert Root-Bernstein. How is this possibly relevant?
- Response:
The reader must be able to judge whether any conflicts of interests may have influenced the results.
This manuscript is a resubmission of an earlier submission. The following is a list of the peer review reports and author responses from that submission.
Round 1
Reviewer 1 Report
The authors describe the binding of morphine and a series of ligands for different neurotransmitter receptors characterised by the presence to cysteines and/or methionines, to glutathione, thus suggesting the possibility of functional interactions among the receptors, their ligands, morphine and glutathione, with consequences to be considered during anesthesia, analgesia etc. This is a mostly empirical and in vitro only work, although important as dissect important interactions that have been reported in previous in vivo/ex vivo work.
I have few suggestions and one major concern.
Abstract:
Line 13, it will be good to add ‘these “……Some cysteine-containing peptides derived from THESE receptors do ……..”
It may be helpful if the authors could add a scheme of the possible interactions/no interactions, since they tested so many ligands and regions from different receptors.
Finally, there are no error bars in the graphs and no reported statistical analysis(es), how many times were the ‘binding assays’ repeated and confirmed?
Reviewer 2 Report
In this manuscript the authors used ultraviolet spectrophotometry to study interactions of glutathione and glutathione-like sequences of opioid and aminergic receptors with ascorbic acid, adrenergic and opioid drugs, and suggested that these interactions may provide novel insights into the impact of adrenergic, opioid and antioxidant systems interactions on anesthesia and addiction. The finding is interesting. However, several concerns significantly dampen enthusiasm.
- The method of ultraviolet spectrophotometry was not explained well. For example, it’s unclear what concentration of opioids and ascorbic acid was used against different concentrations of glutathione in Fig 1. Were all at 10 uM? If yes, how was this concentration determined?
- Also, the assay was performed in sodium phosphate buffer, pH 7.0. It’s unclear that the buffer conditions such salt concentrations and pH have been optimized. A buffer system mimicking physiological conditions such as extra or intracellular compartment should be considered.
- It’s unclear how Kd values were calculated. It seemed only two replicates. It’s unclear how variable of the replicates was. The replicate data on each figure should be shown.
- The authors examined several receptor regions. However, some of these regions share high homology with other receptors or proteins, raising questions regarding their specificity.
- Using a region of a GPCR to test its binding profile with other molecules is problematic since it’s out of the context of whole receptor structure and configuration. It’s unclear if it’s physiologically relevant. No any discussion. I wonder if the authors can examine the interactions using whole receptors and compete with designated peptides with molecules.
- Also, there is no any mechanistic study. For example, it would be straightforward to make a mutant peptide, such as a singly cysteine residue mutation, to look into the binding motif.
- In Figs 3-6, the peptides derived from several receptors were used. However, the concentration of the peptides was much higher than those of natively expressed receptors, e.g. opioid receptors are usually at pmol level, raising questions whether the assays are physiologically relevant.